# Formulation Characterization and Pharmacokinetic Evaluation of Amorphous Solid Dispersions of Dasatinib

**DOI:** 10.3390/pharmaceutics14112450

**Published:** 2022-11-13

**Authors:** Sathish Dharani, Eman M. Mohamed, Tahir Khuroo, Ziyaur Rahman, Mansoor A. Khan

**Affiliations:** Irma Lerma Rangel School of Pharmacy, Texas A&M Health Science Center, Texas A&M University, College Station, TX 77843, USA

**Keywords:** dasatinib, amorphous solid dispersion, dissolution, stability, oral bioavailability

## Abstract

The aim of this study was to improve the physicochemical properties and oral bioavailability of dasatinib (DST) by the amorphous solid dispersion (ASD) approach using cellulose acetate butyrate (CAB) as a carrier. Various formulations of ASD (DST:CAB 1:1 to 1:5) were prepared by the solvent evaporation method. ASDs were characterized for physicochemical attributes, stability and pharmacokinetics. Scanning electron microscopy, Fourier transformed infrared, X-ray powder diffraction, and differential scanning calorimetry confirmed the transformation of the crystalline drug into amorphous phase. ASD formation resulted in a 3.7–4.9 fold increase in dissolution compared to DST or physical mixture. The ASDs formulation exhibited relative stability against transformation from the unstable amorphous phase to a stable crystalline phase that was indicated by spectral and X-ray powder diffraction data, and insignificant (*p* > 0.05) decrease in dissolution. T_max_, C_max_ and AUC_0-∞_ of ASD were 4.3-fold faster and 2.0 and 1.5 fold higher than the corresponding physical mixture. In conclusion, the ASD of DST significantly improved dissolution and oral bioavailability which may be translated into a reduction in dose and adverse events.

## 1. Introduction

Dasatinib (DST) is a tyrosine kinase inhibitor [1]. FDA approved it for clinical use in 2006 for adult and children patients [2]. The agency approved indications are Philadelphia chromosome-positive (Ph+) chronic myeloid leukemia (CML) in chronic phase, chronic, accelerated, or myeloid or lymphoid blast phase Ph+ CML with resistance or intolerance to prior therapy including imatinib, and Ph+ acute lymphoblastic leukemia with resistance or intolerance to prior therapy [2]. Additionally, it is considered a first line therapy for CML treatment [3,4], and is available as a coated tablet in various strengths. In animals (mice, rats, dogs and monkeys), as well as in humans, the drug is rapidly absorbed after administration. However, oral bioavailability of DST is rather low, ranging from 14–34% [5]. Moreover, the bioavailability of dispersed tablets in pediatric patients was estimated to be 36% lower than that of intact tablets [2]. The reason for the poor bioavailability of DST may be its poor solubility in aqueous medium, as it is classified as BCS class II. It is widely reported that the bioavailability of BCS class II drugs are solubility and dissolution dependent [6].

Pharmaceutical technologies reported in the literature to increase solubility and dissolution of poorly soluble drugs are complexation with cyclodextrin [7], self- emulsification drug delivery systems [8,9], amorphous solid dispersion (ASD) [10,11], nanocrystals [12,13], and co-crystals [14], etc. ASD is widely used to enhance the dissolution and bioavailability of BCS class II drugs [10,15]. This technology has enabled the realization of the therapeutic potential of many poorly water-soluble drugs, such asvemurafenib, tacrolimus, regorafenib, everolimus, venetoclax, and olaparib [16,17]. In ASD, the crystalline drug is transformed into an amorphous form and stabilized to some extent by polymer(s). Generally, the amorphous form of a drug has high solubility, dissolution rate and bioavailability compared to the crystalline form. Retaining the amorphous form of the drug in the ASD is a major concern, as reversion to the crystalline form jeopardizes the primary objective of improving the dissolution and bioavailability. Hence, polymers play a crucial role in stabilization of the ASDs by preventing the amorphous drug conversion to the crystalline form during the processing and storage of the dosage form [18]. Cellulose derivatives have always been the polymers of first choice for stabilization purposes of amorphous forms of drugs [19]. ASDs can be manufactured by hot-melt extrusion, spray drying [20], solvent evaporation, and the solvent-antisolvent method [21]. Various crystalline and amorphous forms of DST have been reported [22,23]. For instance, the patent application WO 2017/108605 Al has described preparation of the ASD using polyvinylpyrrolidone (PVP), copovidone and hydroxypropyl cellulose [24]. US patent 9,249,134 B2 described the preparation of the ASD using PVP K-30 and hydroxypropyl methylcellulose acetate succinate by complex chemical reactions and grinding techniques [25]. The focus of this paper was to prepare the ASD of DST by the solvent evaporation technique using cellulose acetate butyrate (CAB) as a polymer carrier to increase dissolution and oral bioavailability. This polymer has not been investigated for ASD of DST. Generally, CAB is used as a coating material (up to 10%) and sustained release excipient (30–60%) [26]. Additionally, compared to other polymers (PVP), CAB has a low moisture absorption capacity that is required to retard the crystallization of the drug in the ASD [27]. The ASDs were characterized for physicochemical attributes by destructive and non-destructive methods, stability and bioavailability in New Zealand rabbits.

## 2. Materials and Methods

### 2.1. Materials

DST monohydrate (purity 99.0%) was obtained from Hangzhou Yuhao Chemical Technology, China. CAB 171-15 was obtained from the Eastman Chemical Company (Kingsport, TN, USA). Acetone, sodium lauryl sulfate (SLS), and HPLC grade acetonitrile (ACN), tetrahydrofuran (THF), formic acid and dibasic potassium phosphate were purchased from Fisher Scientific (Asheville, NC, USA). Deuterated DST (DST-*d*_8_) was obtained from Toronto Research Chemicals, Ontario, Canada. Heparinized rabbit plasma was purchased from BioChemed Services, Winchester, VA, USA. All chemicals were of analytical grade or better.

### 2.2. Methods

#### Preparation of Amorphous Solid Dispersions

ASD formulations composition ranging from 1:1 (ASD-1) to 1:5 (ASD-5) weight ratio of drug to polymer (Table 1) were prepared by the solvent evaporation method as described in the literature with some modification [28,29,30,31]. Briefly, 1 g of DST was accurately weighed and dissolved in 200 mL of acetone with the help of sonication, followed by the addition of 20 mL of 0.5% aqueous SLS solution to further aid in drug solubilization. The polymer (CAB) was gradually added to the drug solution followed by sonication for 10 min. Acetone was evaporated under reduced pressure using a vacuum dryer at 40 °C. The ASDs were powdered in a mortar and pestle and sieved through sieve #60. The particle size fraction that passed through sieve 60 was used in characterization studies. Physical mixtures (PM-1, PM-2, PM-3, PM-4 and PM-5) with a composition identical to the corresponding ASD were prepared by physical mixing of the drug and the polymer.

### 2.3. Morphology

The powder morphology of the drug, polymer, physical mixture and ASDs was visualized by scanning electron microscopy (SEM, JSM-7500F; JEOL, Tokyo, Japan). The samples were approximately 5 nm and coated with carbon using a sputter coater (Cressington, 208 HR with MTM-20 High Resolution Thickness Controller) under high vacuum (argon gas pressure 0.01 mbar) and a high voltage of 40 mV. Morphology was captured at a working distance of 15 mm, an accelerated voltage of 5 KV, and an emission current of 20 µA.

### 2.4. X-ray Powder Diffraction

The X-ray powder diffraction (XRPD) patterns of the samples were collected using a Bruker D2 Phaser SSD 160 diffractometer (Bruker AXS, Madison, WI) equipped with the LYNXEYE scintillation detector and Cu Kα radiation (*λ* = 1.54184 Å). Weighed powder (500 mg) was placed in a sample holder and scanned over 2θ range of 5–30° at a voltage of 30 KV and a current of 10 mA with a step size of 0.0202° at 1 s per step (1733 total steps). Samples were rotated at 15 rpm during measurements to collect the average diffractogram of the samples. The XRPD data were collected using Diffract.Suite^TM^ V4.2.1 (Bruker AXS, Madison, WI, USA).

### 2.5. Fourier Transformed Infrared Spectroscopy

Fourier transformed infrared (FTIR) spectra of the samples were collected using modular Nicolet^TM^ iS^TM^ 50 fitted with ATR accessories (Thermo Fisher Scientific, Austin, TX). Data was collected by placing a small quantity of the sample over the diamond crystal (ATR mode) and pressed with the attached arm to remove air in the sample. The FTIR data collection parameters were a scanning range 400–4000 cm^−1^ with a resolution of 8 cm^−1^ and 32 scans. 

### 2.6. Near-Infrared Spectroscopy

The near infrared (NIR) data of the samples was generated by using the modular Nicolet™ iS™ 50 system (Thermo Fisher Scientific, Austin, TX). The instrument was equipped with a scanning grating monochromator and a diffuse reflectance apparatus (rapid content analyzer). NIR spectra ranging from 4000 to 10,000 cm^−1^ with a data resolution of 8 cm^−1^ and 32 scans were collected after conducting the diagnostic and reflectance tests. After the instrument passed the diagnostic tests and reflectance standardization, samples in 20 mL borosilicate glass vials were placed on the sample window and centered with an iris. Spectral acquisition was performed with OMNIC software, version 9.0 (Thermo Fisher Scientific, Austin, TX, USA).

### 2.7. Differential Scanning Calorimetry

Thermograms of the samples were collected using differential scanning calorimetry (DSC) (Q2000, TA Instruments Co., New Castle, DE, USA). The sample equivalent to 1–2 mg was hermetically sealed in an aluminum pan. The sample was heated to 350 °C at 10 °C/min. This temperature range would cover the melting point of the ASDs components. The nitrogen gas flowed at a pressure of 20 psi to provide inert atmosphere during measurement.

### 2.8. Near-Infrared (NIR) Hyperspectroscopy

To determine the distribution of DST in the ASDs, hyperspectral images of the samples were collected using a Via-Spec II Hyperspectral Imaging system. The instrument details and method of analysis were described in our earlier publications [6]. The model was developed using physical mixtures containing 0–100% DST. The data acquisition software used was Middleton Spectral Vision (Middleton Spectral Vision, Middleton, WI, USA), and the data analysis software was Prediktera Evince™ (Prediktera AB, Umea, Sweden).

### 2.9. Assay

An assay of the formulations was performed by the content uniformity method. The ASD formulations equivalent to 40 mg of DST were transferred into a 100 mL volumetric flask containing the mobile phase. The samples were sonicated for 30 min, filtered using 10 µm filter disso (Sun-Sri, Telford, PA, USA) and injected into the high-performance liquid chromatography (HPLC) to determine DST.

### 2.10. Dissolution 

Dissolution of the formulations was performed using USP Apparatus 2 (Agilent 708-DS, Santa Clara, CA, USA) equipped with an auto sampler (Agilent 850-DS Dissolution Sampling Station). The ASD capsules containing 40 mg of DST (size 00) were inserted into a sinker and placed in the dissolution vessel. The US Food and Drug Administration (USFDA) recommended dissolution method is 1000 mL acetate buffer (pH 4) containing 1% Triton X-100 at 60 rpm, paddle and sampling points 10, 15, 30 and 45 min. The FDA method was modified as 900 mL acetate buffer (pH 4) containing 1% *w*/*w* SLS at 37 °C and 60 rpm. One mL samples were collected at 5, 15, 30, 45 and 60 min and analyzed by the validated HPLC method. A dissolution experiment was performed in triplicate. A two-tailed t-test with a significance level (p) of 0.05 was used for statistical comparison of dissolution profiles of all formulations.

### 2.11. High Performance Liquid Chromatography

The HPLC system used was a 1260 Infinity Series (Agilent technologies, Santa Clara, CA, USA) consisting of a quaternary pump, an automatic injector, a variable wavelength detector, and a column oven. Separation was achieved on a SunFire^TM^ C18 column (150 mm × 4.6 mm, 5 µm) fitted with a Phenomenex guard column (3.9 × 20 mm) and maintained at 30 °C. The mobile phase consists of ACN (A), THF (B) and 10 mM phosphate buffer pH 5.5 (C). A linear gradient elution with a flow rate of 1.0 mL/min was used. The gradient elution was as follows: 0–3 min 20–75% mobile phase A, 5% mobile phase B, 75–20% mobile phase C, 3–7 min 75–20% mobile phase A, 5% mobile phase B, 20–75% mobile phase C, and 7–10 min 20% mobile phase A, 5% mobile phase B, and 75% mobile phase C. The injection volume was 20 µL and the DST was detected at 323 nm. Data was collected using Agilent Open Lab CDS software. The analytical method was validated as per the International Conference of Harmonization guidelines [32].

### 2.12. Stability Study

The short term stability of two ASDs (ASD-1 and ASD-5) were performed by transferring to 20 mL scintillation vials and stored at 40 °C/75% RH for one week in a humidity chamber (KBF720; Binder Inc., Bohemia, NY, USA). After one week of exposure, ASDs were characterized for physicochemical transformation, if there were any.

### 2.13. Pharmacokinetics 

Albino New Zealand white rabbits (n = 4, male:female 1:1) weighing 2–4 kg were used for comparative pharmacokinetic study. ASD-5 and corresponding PM-5 was used as formulations in the study. Food and water were provided ad libitum. The study was performed as per the protocol #IACUC 2020-0046 and approved by Texas A&M’s Institutional Animal Care and Use Committee. The ASD-5 or PM-5 tablets (equivalent to 10 mg of DST) were administered as a single oral dose to the animal using a *pill popper (Emoly pet pill disperser)*. A parallel design was used with a washout period of 21 days. Blood samples (1 mL) were collected from the central or marginal ear vein at 0, 0.25, 0.5, 1, 2, 3, 4, 6, 8 12, 24, 36, 48 and 72 h. The animals were administered acepromazine 0.1–1 mg/Kg/day intramuscular prior to blood collection. The collected blood samples were centrifuged and the plasma was stored at −80 °C before processing for Ultra-Performance liquid Chromatography-Mass spectrometry (UPLC-MS) analysis. 

The plasma samples (300 µL) were transferred to a 2 mL Eppendorf tube followed by the addition of 900 µL methanol, 100 µL DST-*d8* (internal standard) and 100 µL ammonium formate (5 mM, pH 4.0). The samples were vortexed for 1 min and centrifuged at 13,300 rpm and 4 °C for 15 min. The supernatant was analyzed by UPLC-MS for DST quantification.

### 2.14. UPLC-MS Analysis

A validated UPLC-MS method was used for quantification of DST in the rabbits’ plasma. An ACQUITY™ QDA (Waters Corporation, Milford, MA, USA) mass detector was operated in positive ion mode with the following conditions: nitrogen gas flow 100 psi, probe temperature 350 °C, capillary voltage 0.8 KV, cone voltage 5 V, and scan duration 0–4 min. Separation was performed on a Poroshell 120 EC-C18 (4.6 × 50 mm, 2.7 μm), maintained at 30 °C. The isocratic elution method was used with mobile phase consisting of 5 mM ammonium formate (pH 4.0, adjusted with formic acid) and ACN (35:65, *v*/*v*). The flow rate was 0.5 mL/min with an injection volume of 10 µL, and the run time was 5 min. The retention times of DST and DST-*d8* were 2.20 and 2.24 min, respectively, and their peaks were well separated from the baseline. DST and DST-*d8* were detected at *m*/*z* 488 and 496, respectively. Calibration curve concentration ranged from 15 to 200 ng/mL. Precision and accuracy of the method were determined using 15, 45, 100 and 200 ng/mL samples and met the requirement of ±15% of nominal concentration. The method was validated as per FDA bioanalytical method validation guidance [33]. 

## 3. Results and Discussion

### 3.1. ASD Characterizations

Initial and stability samples of the ASD formulations were characterized by SEM, XRPD, FTIR, NIR, DSC, and NIR hyperspectroscopy and dissolution. However, FTIR, SEM, XRPD, NIR and DSC data of ASD-1 and ASD-5 presented in the figures as these formulations represented extreme of drug to polymer composition studied in this paper. However, NIR hyperspectroscopy and dissolution of all the formulations was provided. The paper discussion was focused on ASD-1 and ASD-5 because they represented extreme end of the drug to polymer composition. Furthermore, these formulations were tested in the stability study.

#### 3.1.1. Morphology

Scanning electron micrographs of individual component, PM and ASDs are shown in Figure 1. The DST powder showed plate shaped crystals interspersed with hexagonal shaped crystals. CAB appeared as a lump mass with perforated rough surfaces and crevices. The physical mixture showed the crystals of DST and lumps of CAB. On the other hand, the ASDs powder did not show characteristic shaped crystals of DST or lumps of CAB that may suggest phase transformation. No significant morphological changes were observed in ASD-5 stored at 40 °C/75% RH compared to the initial sample (Figure 1). 

#### 3.1.2. X-ray Powder Diffraction

XRPD patterns of DST, CAB, physical mixtures and ASDs are shown in Figure 2 The crystalline nature of DST was indicated by sharp reflection peaks at 2θ values of 9.0, 11.0, 12.2, 13.7, 14.5, 15.1, 16.1, 17.8, 18.3, 19.0, 19.4, 21.1, 21.6, 22.1, 23.0, 23.4, 24.3, 24.9, 25.7, 27.8, 28.5 and 28.9°. The CAB exhibited an amorphous halo pattern. The PMs showed low intensity DST peaks due to the dilution with the polymer. In the case of ASD-1, it showed a multiple crystalline shape that was not present in either the drug or PM. These peaks maybe due to new polymorphic forms of the drug. To confirm this hypothesis, the DST was recrystallized in acetone. The peaks in ASD-1 matches with the diffractogram of the recrystallized DST. Similar findings were observed in the case of other ASDs (ASD-2, ASD-3, ASD-4 and ASD-5), however the peak intensity was significantly reduced. This indicated that the drug was not completely transformed from crystalline to amorphous form in the ASDs (Figure 2). This suggested that major drug fractions present in the ASD were amorphous. 

In stability condition exposed samples, low grade crystalline conversion in the ASD-5 was noticed when compared to ASD-1. Thus, the degree of crystalline reversion was dependent on the drug to polymer ratio in the formulation when exposed to high temperature and humidity. ASD-1 exposed to the stability condition showed reflection peaks of the recrystallized drug at 9.0, 11.0, 13.3, 19.0, 19.4, 24.6, and 25.7°, but of low intensity compared to the PM-1 (Figure 2). The ASD-5 also showed very low intensity reflection peaks of the recrystallized drug at 6.6, 13.7, 14.5, and 17.8°, which were not present in the initial sample (Figure 2). In both of the samples, the original DST reflection peaks were also noted at 12.2 and 14.5°. Thus, the amorphous drug crystallized to various polymorphic forms on exposure to 40 °C/75% RH. Additionally, recrystallization was more pronounced in ASD-1 compared to ASD-5. This was due to fact that ASD-5 contained higher polymers in the composition than ASD-1 that would increase the glass transition temperature (Tg, Table 1). The Tg of ASDs was calculated using the Gordon-Taylor equation [34,35]. The estimated glass transition temperature of ASD-5 (243 °C) was higher than ASD-1 (217 °C) (Table 1). Thus, it resulted in a decrease in the amorphous to crystalline conversion propensity of the ASD-5. Browne et al. explained the role of the carrier in preventing the recrystallization of a drug in stable ASDs [36]. An increase in Tg of the polymer has been suggested to increase the Tg of the ASD, which may increase the stability of the ASD during storage [36,37]. The physical stability is largely influenced by composition, e.g., polymer and drug characteristics and storage conditions. It is generally known that increasing the storage temperature leads to enhanced crystallization kinetics, and also influences the extent of drug-polymer interactions. It is well known that amorphous materials tend to absorb moisture that reduces the Tg, thereby increasing the molecular mobility and the crystallization tendency of drugs. Nirali et al., observed that even as low as 1% *w*/*w* moisture could cause a major decrease in the Tg of polyvinylpyrrolidone [34]. As expected, a higher moisture content was absorbed in stability exposed ASD-1 compared to ASD-5 due to low polymer content that may further lower the Tg and increase crystallization in the ASD-1. The XRPD results of ASD-5 concurred with literature data representing a stable formulation due to high polymer content compared to other formulations [36,37].

#### 3.1.3. Fourier Transformed Infrared Spectroscopy

Figure 3 shows the FTIR spectra of the individual components, PMs and ASDs. FTIR spectrum of DST showed principal vibration bands at 3458 cm^−1^ (N-H stretch), 3205 cm^−1^ (O-H stretch), 2956 cm^−1^ (methyl C-H stretch), 2886 cm^−1^ (methine C-H stretch), 2850 and 2822 cm^−1^ (methylene C-H stretch), 1681(low intensity) and 1610 (high intensity) cm^−1^ (C=O stretch), 1582, 1544, 1486 and 1461 cm^−1^ (-C-H, HC=CH aryl starches), 1054 (primary alcohol C-O stretch), and 739 and 712 cm^−1^ (C-Cl stretch). Apart from these bands some triplet (1303, 1289 and 1268 cm^−1^; 1199, 1183 and 1167 cm^−1^) and doublet bands (1132, 1120 cm^−1^; 998, 982 cm^−1^; 913, 906 cm^−1^) were also appeared. Unlike XRPD, DST and recrystallized DST did not show any spectral differences. Spectra of PMs and ASDs showed characteristic bands of DST and CAB. However, spectra of the ASDs showed many differences compared to PMs. The carbonyl band of DST in ASDs at 1681 cm^−1^ disappeared while the band at 1610 cm^−1^ moved to 1618 cm^−1^. Bands due to N-H and O-H stretching vibration at 3458 cm^−1^ and 3205 cm^−1^ disappeared, but appeared as broad bands in the range 3100–3133 cm^−1^. (O-H stretch). These changes in the ASDs spectra may be due to hydrogen interactions and hydrophobic interactions between the drug and the polymer [38,39]. Furthermore, these changes in spectra in ASDs may be due to phase transformation of the drug from crystalline to amorphous. 

Some of the ASD formulations showed significant changes in the spectra while others showed insignificant changes after storage. FTIR spectra of stability exposed ASD-1 indicated the appearance of a DST peak at 3458, 3205, 1681, 1610 cm^−1^ that was disappeared or appeared as notch in the initial samples (Figure 3). This indicated the crystalline reversion of the amorphous drug into crystalline form on exposure to a high humidity and temperature condition [40,41,42,43]. On the other hand, ASD-5 did not show major changes in the spectra compared to previously exposed samples. This indicated that the ASD-5 may be more stable than ASD-1.

#### 3.1.4. Near-Infrared Spectroscopy

The NIR spectra of DST, polymer, PMs and ASDs showed broad and overlapping bands due to vibrations of fundamental functional groups such as N-H, C-H, O-H, C-O and C-C [34,44]. The spectra of DST displayed characteristic bands at 8817, 8743 (2nd over tone, C-H stretching), 7366, 7181, 6726 (1st over tone O-H stretching), 6317, 6140, 6086, 5955 (1st over tone, C-H stretching), 5947, 5905 (1st over tone, C-H stretching), 5137, 4886, 4767 (O-H stretching), 4632, 4528, 4466, 4393 (O-H, C-C and C-H stretching), 4323, 4269 (C-H stretching), 4157, 4134, 4080 cm^−1^, doublet peaks at 4200, 4215 cm^−1^, 5804, 5770 cm^−1^; 6009, 6032 cm^−1^, and triplet peaks at 4651, 4674 and 4705 cm^−1^. On the other hand, CAB showed absorption bands at 7050, 5943, 5820, 5245, and 4693 cm^−1^. The PMs showed additive spectra encompassing characteristics bands of both components. For example, a doublet appeared at 5133 and 5245 cm^−1^ in PM-1 due to equal contribution of DST and CAB bands respectively. However, the intensity of the DST bands at 8817, 8743, 7366, 7181, 6726, and 6317 cm^−1^ gradually decreased as the proportion of the drug decreased with respect to the CAB in the PMs. Similar to FTIR, the NIR spectra of ASDs showed many differences compared with PMs. DST bands at 4393, 4767, 4886, and 5137 in ASDs disappeared. There were many other changes in slope and intercept of spectra that may suggest the drug transformation into amorphous form (Figure 4). 

Similar to FTIR, changes were observed in the NIR spectra of the ASDs after exposure to stability conditions. Significant changes were observed in ASD-1 compared to ASD-5 (Figure 4). The spectra of stability exposed ASD-1 resembled PM-1 with the appearance of the DST peaks at 4393, 4767, 4886, and 5137, albeit of low intensity compared to physical PM-1. This indicated the crystalline reversion of the amorphous form. The crystalline reversion decreased as the polymer percentage in the formulation increased. No changes or insignificant changes were observed in the ASD-5. This could be due to the presence of a higher percentage of the polymer that prevented the amorphous to crystalline conversion. Data of FTIR and NIR concurred with XRPD data in terms of the amorphous to crystalline conversion in ASD-1. 

#### 3.1.5. Differential Scanning Calorimetry

The DST showed a melting endothermic peak at 280 °C (Figure 5) that concurred with the crystalline nature of the drug [22]. The drug also showed a broad peak at 135 °C starting above 100 °C that could be due to the water molecule associated with the crystalline structure. PM-1 showed both peaks, albeit of low intensity, due to the presence of the polymer. On the other hand, increasing the polymer percentage in the formulation resulted in complete disappearance of the drug peaks (PM-5 did not show any peak). The thermogram of ASD-1 showed a broad peak at 140 °C but not the drug melting peak. This may be due to water loss from the crystalline lattice or a new polymorphic form of the DST. This indicated that the drug was partially transformed into the amorphous form. The crystalline fraction present in the ASD-1 may be the new polymorphic form. These observations concurred with the XRPD data. On the other hand, ASD-5 did not show any of the thermal event peaks. This may be due to the drug conversion to amorphous form or the crystalline fraction present being too low to appear as a peak in the thermogram as supported by the PM-5. 

Changes were observed in thermograms of the ASD-1 stability condition exposed sample compared to its initial sample. A very low intensity peak at 170 °C and a broad peak at 260 °C appeared in the ASD-1 in addition to the 140 °C peak. The peak at 260 °C could be due to the melting of DST that was depressed by the polymer or the new polymorphic form [45]. These changes indicated an increase in crystalline fraction and polymorphic transformation in ASD-1 stability samples. In the case of the ASD-5 stability sample, a broad and a low intensity peak at 240 °C was noticed that was not present in the initial sample. This peak may be the melting peak of the DST that was suppressed by the polymer or new polymorphic form. This observation was supported by FTIR and XRPD data indicating insignificant phase transformation in ASD-5 (Figure 5).

#### 3.1.6. Near-Infrared Hyperspectroscopy

NIR hyperspectroscopy is a nondestructive method that provides spectral and spatial distribution of formulation components. The data was mathematically pretreated using mean centering, and standard normal variate methodologies were used to normalize and correct the baseline deviation. The partial least square (PLS) concentration images of samples were created using DST or CAB as library components or predictors [6]. Initially, the PLS model was developed using known concentrations of DST and CAB in different ratios from zero to 100% of the DST. The yellowish-orange and blue pixels indicated high and low concentration of DST in the sample matrices if DST was used as a predictor. Figure 6 shows the distribution of the DST in ASD samples from ASD-1 to ASD-5. As the amount of CAB increased in ASDs, the pixel color changed from yellowish-orange to blue. The reverse trend was observed when CAB was used as a predictor. Furthermore, pixel color distribution was uniform, which indicated uniform and or molecular distribution of the components in the ASDs. A quantitative analysis was also performed using a PLS calibration plot. The amount of DST present in formulations ASD-1, ASD-2, ASD-3, ASD-4 and ASD-5 was 48.0 ± 3.6, 31.9 ± 2.8, 24.6 ± 3.2, 19.4 ± 2.5 and 16.3 ± 2.4% of nominal values, respectively. 

The PLS concentration images did not show any change in both color and distribution of the pixels in ASD-1 and ASD-5 formulations after exposure to stability condition for a month. Furthermore, a quantitative analysis indicated no significant change in the amount of the drug in the ASDs. DST present in initial and exposed ASD-1 and ASD-5 was 48.0 ± 3.6, 46.3 ± 2.9% and 16.3 ± 2.4, 17.4 ± 3.3% of nominal values, respectively. This indicated no significant degradation of the drug in the ASDs during the stability period (Figure 6).

#### 3.1.7. Dissolution 

The drug content in the ASDs ranged from 95.8 ± 0.9 to 98.2 ± 1.2% of nominal values. The ASDs formulation resulted in a significant increase in the DST dissolution when compared to DST or corresponding PM. No significant difference in dissolution between the drug and its physical mixtures were observed that indicated that the polymer did not increase dissolution per se. The dissolution of DST from PM-1, PM-2, PM-3, PM-4 and PM-5 were 18.2 ± 4.8, 23.8 ± 3.6, 20.9 ± 2.8, 19.9 ± 3.1, 19.7 ± 2.6 and 20.1 ± 4.2% in 60 min, respectively (Figure 7). Moreover, dissolution increased significantly in the ASD formulations. This was due to the transformation of the crystalline drug into the amorphous from in the ASDs. Furthermore, dissolution increased with an increase in the polymer proportional to the drug in the formulation. This was due to the higher fraction of the amorphous form as the polymer percentage in the ASD formulation increased. However, an increase in the dissolution was not proportional to polymer content in the ASD. The highest increase in the dissolution was observed in ASD-4 and ASD-5. Theses formulations contained 80% and 83.3% polymer content, respectively. The dissolved drug was 42.9 ± 0.8, 69.4 ± 0.5, 75.3 ± 0.9, 76.8 ± 0.1, and 88.8 ± 2.7% in 60 min from ASD-1, ASD-2, ASD-3, ASD-4 and ASD-5, respectively (Figure 7).

No statistical difference (*p* > 0.05) in the assay values was observed after exposure to the stability condition. The assay values of ASD-1 and ASD-5 were 96.2 ± 1.4 to 97.6 ± 1.8% after exposure to 40 °C/75% RH. Dissolution profiles of the ASD formulation changed after exposure to stability condition. A decrease in dissolution was observed from 42.9 ± 0.8 to 20.4 ± 4.8%, and 88.8 ± 2.7 to 81.12 ± 1.36% in ASD-1 and ASD-5, respectively (Figure 7). A decrease in dissolution can be explained by crystalline reversion. It is well known that an amorphous drug converts to a crystalline drug when exposed to high humidity and temperature. Crystalline reversion is often manifested by a decrease in dissolution, which is supported by XRPD data. However, a decrease in dissolution was significantly higher in ASD-1 compared to ASD-5. This was due to the higher transformation of amorphous DST to its crystalline form compared to ASD-1, which was corroborated by XRPD, spectroscopy and DSC data. The degree of crystallization in ASD-1 and ASD-5 were related to the polymer percentage in their composition. A higher polymer concentration was accompanied by an increase in glass transition temperature and higher stability of the ASD. ASD-1 and ASD-5 contained 50% and 83.3% polymers in their composition, respectively.

Dissolution profiles of two formulations can be compared by similarity (*f2*) and difference (*f1*) factors. The profiles are considered similar if *f2* > 50 and *f1* < 15, as per FDA guidance (FDA 1997). Based on the *f1* and *f2* values comparison between initial and corresponding exposed formulations, ASD-5 met the similarity criteria. The *f1* and *f2* values were 9.9 and 59.9 in ASD-5, respectively.

#### 3.1.8. Pharmacokinetics

The drug was quantified up to 12 h and detected up to 24 h from both the formulation in the plasma. The pharmacokinetic profiles and parameters of the two formulations look completely different (Figure 8). ASD showed shorter T_max_, and higher C_max_ and AUC_0-∞_ compared to PM. ASD-5 showed faster absorption and gradual elimination while PM-5 showed very slow absorption and faster elimination due to low plasma concentration. These differences can be explained by faster dissolution and attainment of supersaturation in the ASD in the in vivo milieu compared to PM. In vitro data supported that argument, as 20.1 and 88.8 ± 2.7% DST was dissolved in PM-5 and ASD-5, respectively. The increase in dissolution was 4.4-fold in ASD compared to the physical mixture. This was due to amorphous drug transformation in the ASD. Statistically significant differences in pharmacokinetic parameters were observed between ASD-5 and PM-5 (*p* < 0.05). The T_max_, C_max_ and AUC_0-∞_ of ASD-5 and PM-5 were 0.9 ± 0.3 and 3.8 ± 0.5 h, 79.7 ± 8.2 and 40.1 ± 16.5 ng/mL, and 318.4 ± 80.7 and 185.5 ± 63.0 ng/mL·h, respectively. Furthermore, T_max_, C_max_ and AUC_0-∞_ of ASD-5 were 4.3-fold faster, and 2.0 and 1.5 higher than PM-5, respectively. Thus, the bioavailability of ASD was 198.7 and 145.7% when calculated with respect to AUC_0-∞_ and C_max_, respectively compared to PM. A plasma concentration C_max_ of 50 ng/mL is required to maintain clinical response and C_min_ concentration of <2.5 ng/mL is required to avoid pleural effusion [46,47,48]. A C_min_ concentration of 2.5 ± 2.0 and 3.1 ± 0.6 ng/mL was achieved at 24 h in PM and ASD, respectively. ASD-5 maintained the required C_max_ concentration while PM-5 did not achieve therapeutic C_max_ concentration. The ASD formulation has better bioavailability and potentially better therapeutic outcomes compared to PM. It is also conceivable that the dose of DST can be further reduced to obtain bioequivalent products with respect to commercial formulations.

## 4. Conclusions

The conversion of crystalline DST to the amorphous phase was achieved and stabilized to a certain extent by CAB in ASD. Electron microscopy, spectroscopy, X-ray powder diffraction and thermal data confirmed the phase transformation of the crystalline drug. The drug and polymer were uniformly distributed within the matrix, as suggested by hyperspectroscopy images. Higher drug to polymer composition (drug:polymer 1:5) showed a statistically significant (*p* < 0.05) increase in the dissolution (4.4 fold compared to the physical mixture). Moreover, the amorphous drug was relatively stable when the polymer percentage was 83.3% of the composition with no significant change in dissolution rate or extent (*p* > 0.05) after storage. This was due to the high Tg value and low moisture content of the composition. The improvement in pharmacokinetic parameters were noted with the ASD. T_max_, C_max_ and AUC_0-∞_ of ASD were 4.3-fold faster, and 2.0 and 1.5 times higher than PM, respectively. Thus, the ASD formation results in a significant improvement in oral bioavailability (1.5–2.0 fold of PM). The dose of the drug can be reduced significantly based on the current data that may result in the improvement in clinical outcome and reduction in adverse events. However, there is a need for further preclinical studies in animals and clinical studies in humans, and exhaustive stability studies to bring ASD formulations of DST to clinical practice.

## Figures and Tables

**Figure 1 pharmaceutics-14-02450-f001:**
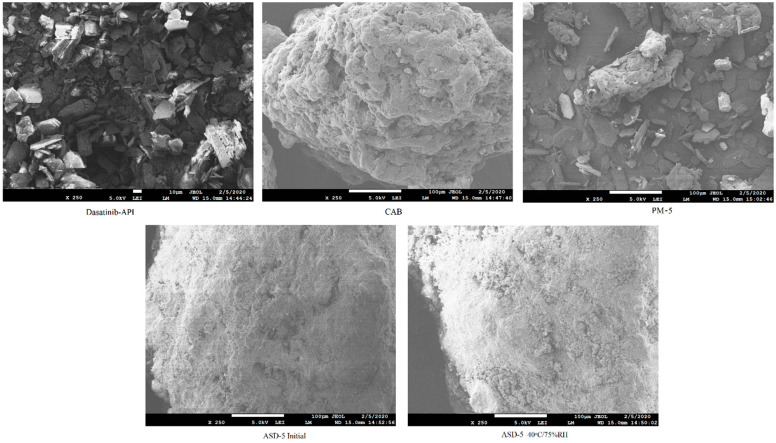
Scanning electron microscopic images of DST, CAB, PM-5, ASD-5 initial and 40 °C/75% RH exposed.

**Figure 2 pharmaceutics-14-02450-f002:**
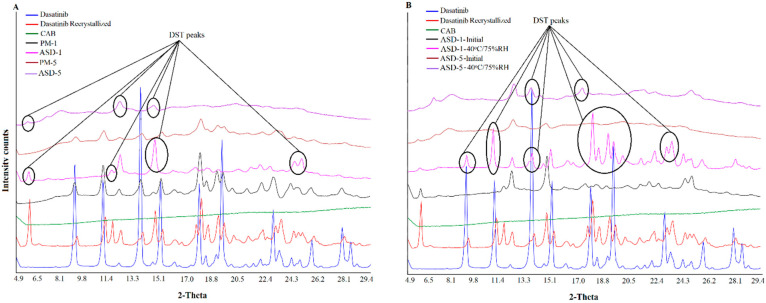
X-ray powder diffractogram of (**A**) PM-1, PM-5 (**B**) ASD-1, ASD-5 initial and 40 °C/75% RH exposed.

**Figure 3 pharmaceutics-14-02450-f003:**
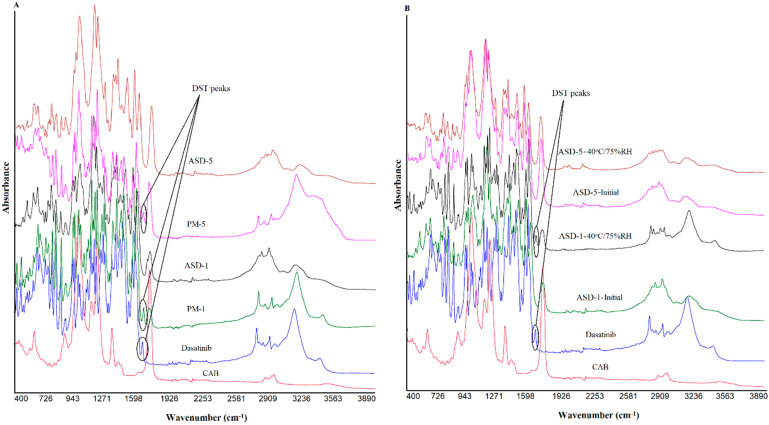
FTIR spectra of (**A**) PM-1, PM-5 (**B**) ASD-1, ASD-5 initial and 40 °C/75% RH exposed.

**Figure 4 pharmaceutics-14-02450-f004:**
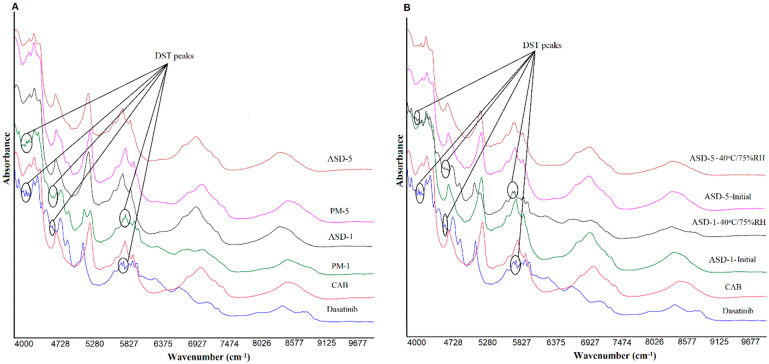
NIR spectra of (**A**) PM-1, PM-5 (**B**) ASD-1, ASD-5 initial and 40 °C/75% RH exposed.

**Figure 5 pharmaceutics-14-02450-f005:**
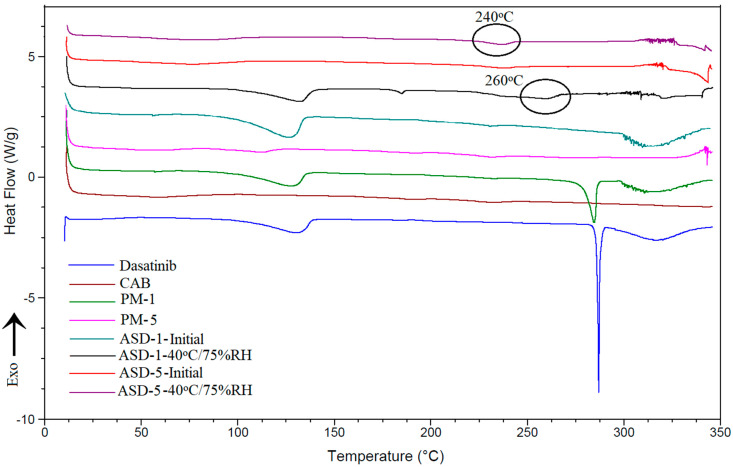
DSC thermograms of PM-1, PM-5, ASD-1, ASD-5 initial and 40 °C/75% RH exposed.

**Figure 6 pharmaceutics-14-02450-f006:**
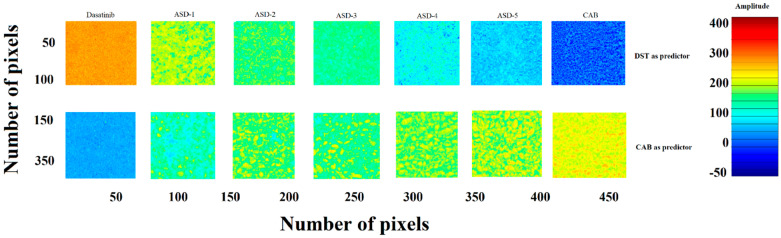
PLS concentration images of ASDs.

**Figure 7 pharmaceutics-14-02450-f007:**
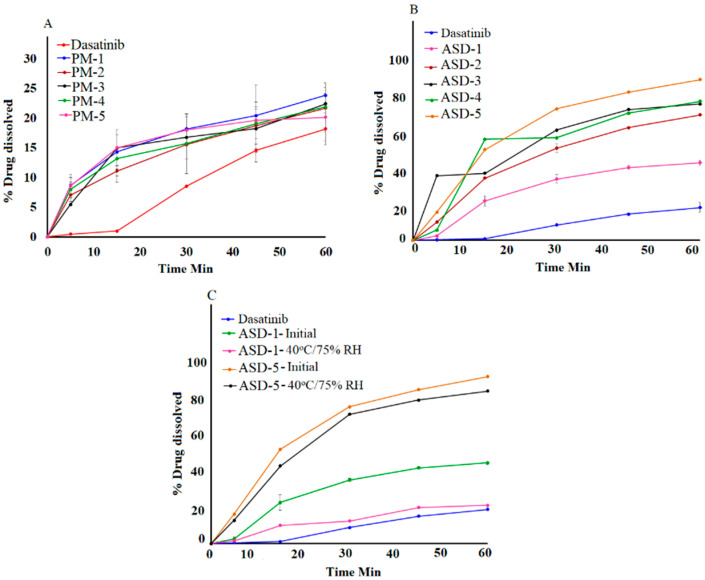
Dissolution profiles of (**A**) Physical mixtures (**B**) ASD-1 to ASD-5 (**C**) ASD-1, ASD-5 initial and 40 °C/75% RH exposed.

**Figure 8 pharmaceutics-14-02450-f008:**
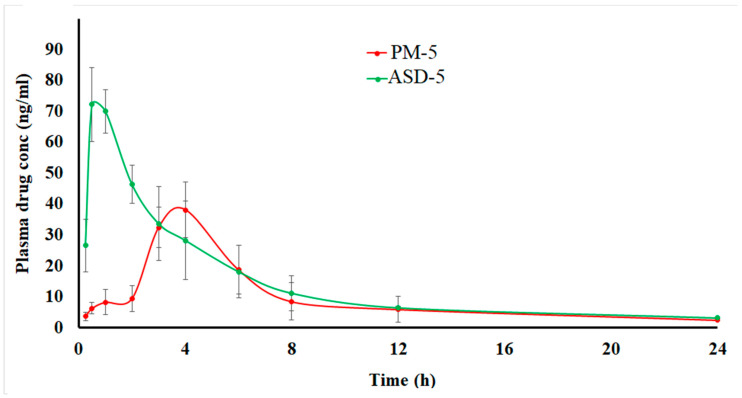
Pharmacokinetic profiles of the PM-5 and ASD-5 compressed tablets in rabbits.

**Table 1 pharmaceutics-14-02450-t001:** Composition of ASDs.

Formulation	% DST	% CAB	Drug to Carrier Ratio	Estimated Glass Transition Temperature (Tg)	Moisture Content (%)
ASD-1	50	50	1:1	217 °C	4.2 ± 0.2
ASD-2	33.3	66.7	1:2	231 °C	3.4 ± 0.1
ASD-3	25	75	1:3	237 °C	2.8 ± 0.2
ASD-4	20	80	1:4	241 °C	2.3 ± 0.3
ASD-5	16.6	83.4	1:5	243 °C	1.8 ± 0.0

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
