# Peer review of "Formulation Characterization and Pharmacokinetic Evaluation of Amorphous Solid Dispersions of Dasatinib"

_pharmaceutics, 2022, doi:10.3390/pharmaceutics14112450_

Round 1
Reviewer 1 Report
The manuscript pharmaceutics-1975353 discussed dasatinib amorphous solid formulations with cellulose acetate butyrate (CAB) that improved dissolution and pharmakinetics compared to their physical mixture, which has great significance in Pharmaceutical field. But I field there are so many loopholes in the results and discussion e.g. ASD-1/5 are not pure amorphous phases, that are converted to crystalline forms during exposure humidity. In that sense, claiming amorphous solid dispersion is correct. I have following comments and the manuscript needs major revision before acceptance.
1. Abstract: Write full form of CAB.
2.0 and 1.5 'fold' higher than
2. Table 1 shows ASD1-5 compositions, but authors showed characterization of ASD 1&5.
3. FTIR have you recorded in ATR mode or KBR pellet?
4. Page 5: Lumps (SEM images) are also observed for CAB and DST-CAB. Then how you can confirm that phase transformation occurs in ASDs?
5. Line 232: Between the drug and the polymer
6. What is the meaning of CO band of DST disappeared from ASD?
7. Line 251: Reason behind DST bands disapperance in ASDs? Can you assign NIR bands?
8. Authors need to check whether anhydrous form (CCDC refcode: RAVPUB) or any other new form is there in ASDs. Need to specifiy.
9. Line 266: There is no discussion on humidity effect on these solid forms,..But Fig 4B is provided. Why authors put stability discussion at the end, when the data are given in characterization part?
10. Figure 5: Need to know the Tg of ASD 1/5 to show the improved stability of ASD 5.
11. Line 310: These values are in % or any unit?
12. Where is your PM-1/5 dissolution data? Why suddenly all ASD 1-5 dissolution data are provided? Need esd for dissolution data.
13. Stability discussion at the end is no use. Better to insert where you provided characterization.
14. Authors should focus on ASD5 (in terms of improved dissolution/pharmacokinetics) than other formulation in the conclusion.
Author Response
Authors would like to thank all the reviewers and editor for their constructive criticism and comments. We have addressed all the concerns in this reply and changes are highlighted in green color in the manuscript.
Reviewer #1:
Comment #1: The manuscript pharmaceutics-1975353 discussed dasatinib amorphous solid formulations with cellulose acetate butyrate (CAB) that improved dissolution and pharmacokinetics compared to their physical mixture, which has great significance in Pharmaceutical field. But I field there are so many loopholes in the results and discussion e.g. ASD-1/5 are not pure amorphous phases, that are converted to crystalline forms during exposure humidity. In that sense, claiming amorphous solid dispersion is correct.
Response: We agree that ASD’s are not completely amorphous. However, major fraction was amorphous especially in ASD-5. Moreover, there are many FDA approved products which are not completely amorphous but still called as ASDs due to the fact that major fraction of the drug is amorphous. Examples is tacrolimus amorphous solid dispersion products.
Comment #2: Abstract: Write full form of CAB. 2.0 and 1.5 'fold' higher than.
Response: We have modified the manuscript as per reviewer suggestion.
Comment #3: Table 1 shows ASD1-5 compositions, but authors showed characterization of ASD 1&5.
Response: We have characterized all ASD formulations. ASD-1 and ASD-5 formulations represented extreme of drug to polymer composition. We provided microscopy, spectral, thermal and X-ray data of ASD-1 and ASD-5 and discussed other ASDs as well. We did not provide data of other ASDs because it would make the figures crowded and would not provide any additional information. Furthermore, we performed stability of ASD-1 and ASD-5 only.
Comment #4: FTIR have you recorded in ATR mode or KBR pellet?
Response: We used ATR mode FTIR. It is clarified in the manuscript.
Comment #5: Page 5: Lumps (SEM images) are also observed for CAB and DST-CAB. Then how you can confirm that phase transformation occurs in ASDs?
Response: Surface morphology of ASD looks completely different form individual components. That may suggest phase transformation. We have modified the sentence in the manuscript.
Comment #6. Line 232: Between the drug and the polymer
Response: Thank you for pointing out type. It is corrected in the manuscript (Line 290).
Comment #7. What is the meaning of CO band of DST disappeared from ASD?
Response: DST showed carbonyl peaks at 1681(low intensity) and 1610 (high intensity) cm− 1. Low intensity peak (1681 cm-1) disappeared in the ASDs possibly due to H-interactions between drug and the polymer as the other characteristics peaks remains intact. This was possibly due to phase transformation during manufacturing.
Comment #8: Line 251: Reason behind DST bands disappearance in ASDs? Can you assign NIR bands?
Response: Disappearance of DST bands at 4393, 4767, 4886, and 5137 cm-1 in ASDs indicated phase transformation compared to physical mixture and API. We have discussed the characteristic peaks of DST in the text and highlighted major changes in figure 3. Since most of the bands were broad and highly-overlapping, as per reviewer suggestion, we could assign only few bands (Line 308).
Comment #9: Authors need to check whether anhydrous form (CCDC refcode: RAVPUB) or any other new form is there in ASDs. Need to specify.
Response: DST used in this work was monohydrate form. There is no water involve in the manufacturing of ASD. So, ASDs would be anhydrous. However, in stability exposed samples, peaks were observed after 100 oC which could be due to water associated with crystalline lattice or new polymorphic form of the DST. It is clarified in the text of the manuscript (Line 338).
Comment #10: Line 266: There is no discussion on humidity effect on these solid forms,..But Fig 4B is provided. Why authors put stability discussion at the end, when the data are given in characterization part?
Response: We have discussed the effect of humidity in Line 265 and modified the discussion part of the manuscript. We have combined initial and stability ‘results and discussion’ section as per reviewer suggestion.
Comment #11: Figure 5: Need to know the Tg of ASD 1/5 to show the improved stability of ASD 5.
Response: Estimated Tg of ASDs temperature is added in the manuscript (Line 254 and Table 1).
Comment #12: Line 310: These values are in % or any unit?
Response: Thank you for pointing out type. It is corrected in the manuscript (Line 390).
Comment #13: Where is your PM-1/5 dissolution data? Why suddenly all ASD 1-5 dissolution data are provided? Need esd for dissolution data.
Response: Dissolution of all PMs was added in Figure 7A and Line 390.
Comment #14. Stability discussion at the end is no use. Better to insert where you provided characterization.
Response: We have combined initial and stability ‘results and discussion’ section as per reviewer suggestion.
Comment 14. Authors should focus on ASD5 (in terms of improved dissolution/pharmacokinetics) than other formulation in the conclusion.
Response: Thank you for a very good suggestion. We have modified the conclusion section of the manuscript.
Reviewer 2 Report
Formulation, Characterization, and Pharmacokinetic Evaluation of Amorphous Solid Dispersions of Dasatinib
Dharani et al.
In the manuscript, the authors describe a new ASD of Dasatinib using cellulose acetate butyrate. The ASDs were characterized before and after stability concerning their solid-state via FTIR, NIR, XRPD and, DSC, and their in-vitro performance was evaluated with dissolution studies. The most promising formulation was then tested in-vivo in pre-clinical species. Although the study presents an extensive amount of work and interesting results the advantage of such ASDs (meaning cellulose acetate butyrate over PVP for example) is not clear and the application and translatability of the results to more relevant industrial processes are not entirely straightforward as the characterization of the ASDs is not done in a way that allows the system to be truly understood. As consequence, the main conclusion of the authors is that more polymer stabilized the API better, a general, not surprising conclusion quite known in the field. Likewise, the reviewer suggests that the manuscript should be accepted only once the authors thoroughly revise the results and discussion and present an understanding of how the polymer-drug interactions will affect the stability and performance of the drug. Only then can be truly understood if this polymer presents an advantage over more well-used ones such as PVP and how the process could be translatable into more industrially relevant processes.
General Comments
Copyright pertains to 2020 and volume 13 of the journal is from 2021.
It is not entirely clear how the use of cellulose acetate butyrate presents an advantage over other more well-used polymers such as PVP, and for which ASDs of DST has already been described.
Moreover, the authors, did not prepare the ASDs via an industrially scalable method such as spray-drying, do the authors think that this could critically impact the performance of the ASDs? Do the authors expect a similar outcome when they would produce the ASDs via spray-drying? Would the miscibility of drug and polymer be expected to be the same? If yes, why?
Moreover, although the authors did a large amount of work to characterize the ASDs it is not always clear why all the techniques were applied, for instance, FTIR and NIR give very similar information, and DSC was applied but no Tg evaluation was done, and although the authors indicate the influence of moisture content on the Tg and stability they did not evaluate this content. Additionally, it is also not clear why only ASD-1 and ASD-5 are characterized for XRPD, DSC, FTIR, and NIR. Likewise, the authors should better explain the rationale behind the application techniques and the objective of the characterization as well as why only certain samples were analyzed, otherwise, the reader might have the impression the techniques were just applied for the sake of applying them.
Overall the organization of the results is confusing, as the reader goes through the text all the characterization results are presented, but then the stability section is just later on, when reading the manuscript first, it did seem like some results description/explanation was missing. The reviewer would suggest that the results are presented all together in the relevant characterization sections. Moreover, if some ASDs were screened fully, and then just some were then chosen for further characterization the manuscript should also follow this rationale, starting with all ASDs and then why some were chosen, and then the results for the most promising formulations.
Lastly, the authors make some conclusions/assumptions without evidence, e.g. the effect of polymer load on the glass transition.
Detailed comments
Introduction
Line 34: “the reason” or “reasons”
Line 36: the bioavailability of BCS class II drugs is
Could the authors please give some examples of how cellulose acetate butyrate is used in pharma? What ranges are normally applied?
Materials and Methods
Line 124: why was modulated DSC not done?
Line 143: is the dissolution study in accordance with compendial methods?
Line 167: the stability study was performed under accelerated conditions, do the authors think one week provides enough evidence about the possible shelf stability of the powders?
Results and Discussion
Line 213: “on the hands”? What do the authors mean?
The reviewer would suggest to first present the XRPD results, as then is clearer what the authors are talking about when they say that in the FTIR and NIR, differences in peaks could indicate the amorphous phase of the drug or a new crystalline phase.
Line 234 and 253: why did the authors not evaluate the recrystallized DST via IR also? Likewise, they would know if the peaks are in line with the new crystalline phase of the drug? Moreoever, the authors could have also produced an amorphous phase of the drug and run it via IR. Likewise, they could have de-coupled the different effects new solid-state phase (amorphous or new polymorph) and OH bonding.
Line 262: Are other DST polymorphic forms known? If yes, could the authors please compare their results with the patterns expected for these forms.
Figure 5: could the authors please indicate the direction of the exo signal.
Line 274: Does DST exist as a hydrate?
Line 277: The authors think that the percentage of DST was so low that it could not be detected?
Line 279: Actually a peak seems t be present at about 260°C, which could be de depressed melting point of the drug actually, moreover also the water peak of DST is present. Do the authors think actually they might have a mixture of a new polymorph with the original one? Also in the XRPD, some peaks of the recrystallized DST are in line with the original polymorph.
Line 281: the reviewer thinks there is a very small peak present at 240°C, again this could be the depressed peak of the original polymorph.
Line 284: Please be more specific about how the DSC results are in line with XRPD ones, how do the results connect? XRPD and DSC are complementary techniques and thus the reviewer thinks that the manuscript would benefit if the authors join the 2 techniques and discuss them together.
Line 284: why did the authors not discuss the results after stability? For instance ASD-1 just the water peak is clearly visible now, why? For ASD-5 the peak at 240°C seems now more preeminent.
Line 284: why did the authors not analyze Tg for the system or the miscibility of the components via melting peak depression in the PM? It would be good to know how the drug is a polymer mix as this is a good predictor of the stability of systems. Moreover, this could also help understand why the polymer did not allow the full amorphization of the drug (poor miscibility some drug-rich zones in the drug could be less stable and crystallinity is promoted, for instance).
Line 302: were these the nominal expected values? Are these within the accepted nominal dose ranges established by compendial guidelines? Moreover, why did the authors here analyze all ASDs and before only ASD-1 and ASD-5?
Line: 324: the authors describe well the effect of moisture uptake on the Tg and stability, thus why was the Tg not evaluated and the moisture content also?
Line 379: how do the authors know that the Tg of ASD-5 was higher? They did not evaluate the Tg via DSC. What is the Tg of the drug, and what is the Tg of the polymer? How do the authors know the polymer has a higher Tg than the drug?
Line 380: please avoid word shortening such as it’s
Line 390: if there was a transformation to the original polymorphic form, why was no peak detected at 280°C?
Line 363: by the analysis of the DSC the reviewer thinks there is a peak at 240°C for the ASD-5 and this peak is more evident after stability, this could appear insignificant, but could indicate already some conversion of ASD-5 after 1 week at accelerated conditions that could explain the mall difference in dissolution, although, for the results of this study this was not very relevant, it could raise some concerns about the long term shelf stability of the formulations and in-use performance.
Line 436: could the authors please add the absolute bioavailability, likewise it would be more straightforward to compare with already marketed formulations that are only 14-34% bioavailable.
Line 437: could the authors please state the Cmin of the Cp v. Time profiles?
Author Response
Authors would like to thank all the reviewers and editor for their constructive criticism and comments. We have addressed all the concerns in this reply and changes are highlighted in green color in the manuscript.
Reviewer #2
Comment #1: In the manuscript, the authors describe a new ASD of Dasatinib using cellulose acetate butyrate. The ASDs were characterized before and after stability concerning their solid-state via FTIR, NIR, XRPD and, DSC, and their in-vitro performance was evaluated with dissolution studies. The most promising formulation was then tested in-vivo in pre-clinical species. Although the study presents an extensive amount of work and interesting results the advantage of such ASDs (meaning cellulose acetate butyrate over PVP for example) is not clear and the application and translatability of the results to more relevant industrial processes are not entirely straightforward as the characterization of the ASDs is not done in a way that allows the system to be truly understood. As consequence, the main conclusion of the authors is that more polymer stabilized the API better, a general, not surprising conclusion quite known in the field. Likewise, the reviewer suggests that the manuscript should be accepted only once the authors thoroughly revise the results and discussion and present an understanding of how the polymer-drug interactions will affect the stability and performance of the drug. Only then can be truly understood if this polymer presents an advantage over more well-used ones such as PVP and how the process could be translatable into more industrially relevant processes.
Response: The focus of the manuscript was not to compare PVP vs CAB as a carrier for ASD formulations of DST but rather to provide an alternative to commonly used carrier. We have provided scientific explanation of physicochemical characterization and improvement of dissolution and pharmacokinetic of the ASD. We have modified the results and discussion as per reviewer suggestions.
Comment #2: Copyright pertains to 2020 and volume 13 of the journal is from 2021. It is not entirely clear how the use of cellulose acetate butyrate presents an advantage over other more well-used polymers such as PVP, and for which ASDs of DST has already been described.
Response: We have not compared comparative physicochemical and pharmacokinetic properties of ASDs of DST prepared using CAB and PVP as carrier. Based on the literature data, PVP is highly hygroscopic, and may promote crystallization in the ASDs compared to CAB due to CAB poor water solubility and hydrophobicity (Line 64). It is expected that CAB may perform better in stability studies compared to PVP as a carrier in ASD formulations.
Comment #3. Moreover, the authors, did not prepare the ASDs via an industrially scalable method such as spray-drying, do the authors think that this could critically impact the performance of the ASDs? Do the authors expect a similar outcome when they would produce the ASDs via spray-drying? Would the miscibility of drug and polymer be expected to be the same? If yes, why?
Response: Our lab is not equipped with spray dryer equipment to prepare ASD of the DST by this method. Based on our previous publications and literature, process and formulation impact performance of ASD formulations due to its impact on amorphous to crystalline fraction in the formulations. We believe that ASD of the DST can be prepared by spray drying process. The miscibility of the drug and polymer may not be same in a solvent system. Further, solvent system has to optimized to implement spray drying process for the ASD of the DST.
Comment #4: Moreover, although the authors did a large amount of work to characterize the ASDs it is not always clear why all the techniques were applied, for instance, FTIR and NIR give very similar information, and DSC was applied but no Tg evaluation was done, and although the authors indicate the influence of moisture content on the Tg and stability they did not evaluate this content. Additionally, it is also not clear why only ASD-1 and ASD-5 are characterized for XRPD, DSC, FTIR, and NIR. Likewise, the authors should better explain the rationale behind the application techniques and the objective of the characterization as well as why only certain samples were analyzed, otherwise, the reader might have the impression the techniques were just applied for the sake of applying them.
Response: FTIR, DSC, and XRPD are orthogonal techniques and commonly used to characterize ASDs formulations. One technique does not provide complete picture of solid phase transformation that is why orthogonal techniques needed to support the findings. We characterize all the ASD formulations (ASD-1 to ASD-5). However, we provided data of ASD-1 and ASD-5 because these formulations represented extreme end of drug to polymer composition. Adding the data of all the ASDs would make the figures crowded and will not provide any new information. Furthermore, we have performed stability of ASD-1 and ASD-5 only.
Comment #5: Overall the organization of the results is confusing, as the reader goes through the text all the characterization results are presented, but then the stability section is just later on, when reading the manuscript first, it did seem like some results description/explanation was missing. The reviewer would suggest that the results are presented all together in the relevant characterization sections. Moreover, if some ASDs were screened fully, and then just some were then chosen for further characterization the manuscript should also follow this rationale, starting with all ASDs and then why some were chosen, and then the results for the most promising formulations.
Response: We have combined initial and stability ‘results and discussion’ section as per reviewer suggestion.
Comment #6: Lastly, the authors make some conclusions/assumptions without evidence, e.g. the effect of polymer load on the glass transition.
Response: We have added estimated Tg of the ASDs in the manuscript (Line 254 and Table 1).
Comment #7: Line 34: “the reason” or “reasons”
Response: Typo corrected in the manuscript (Line 34)
Comment #8: Line 36: the bioavailability of BCS class II drugs is
Response: Typo corrected in the manuscript (Line 36)
Comment #9: Could the authors please give some examples of how cellulose acetate butyrate is used in pharma? What ranges are normally applied?
Response: CAB is used as a coating material (up to 10%) and sustained release excipient (30-60%). Recently, it has gained attention in preparing ASDs due to its low moisture absorption capacity. This information is added in Lines 62-65 of the manuscript.
Comment #10. Line 124: why was modulated DSC not done?
Response: DSC equipment in our is not equipped with modulated option. However, we have estimated Tg using Gordon-Taylor equation (Line 253).
Comment #11: Line 143: is the dissolution study in accordance with compendial methods?
Response: We have used modified FDA dissolution method. However pH of the medium was same. It is stated in the methods section of the manuscript (Line 152).
Comment #12. Line 167: the stability study was performed under accelerated conditions, do the authors think one week provides enough evidence about the possible shelf stability of the powders?
Response: In the industry, preformulation and prototype formulation stability studies are commonly performed by exposing for a week to a month to get sense of stability of the formulation. However, this study does not fully meet ICH/FDA requirements.
Comment #13. Line 213: “on the hands”? What do the authors mean?
Response: Thank you for pointing out typo. It is corrected in the manuscript (Line 221 ).
Comment #14. The reviewer would suggest to first present the XRPD results, as then is clearer what the authors are talking about when they say that in the FTIR and NIR, differences in peaks could indicate the amorphous phase of the drug or a new crystalline phase.
Response: Modified ‘results and discussion’ part as per reviewer suggestion.
Comment #15. Line 234 and 253: why did the authors not evaluate the recrystallized DST via IR also? Likewise, they would know if the peaks are in line with the new crystalline phase of the drug? Moreoever, the authors could have also produced an amorphous phase of the drug and run it via IR. Likewise, they could have de-coupled the different effects new solid-state phase (amorphous or new polymorph) and OH bonding.
Response: FTIR spectra of DST and recrystallized DST did not show differeces in spectra.
Comment #16. Line 262: Are other DST polymorphic forms known? If yes, could the authors please compare their results with the patterns expected for these forms.
Response: Characteristic XRPD peaks for recrystallized DST were mentioned in line 248, and peaks does not match with the literature data of reported polymorphs. Since this work is focused on formulation, crystal structure analysis was not performed. https://www.sciencedirect.com/science/article/pii/S0022354916003245?via%3Dihub
Comment #17. Figure 5: could the authors please indicate the direction of the exo signal.
Response: Figure 5 is modified as per reviewer suggestion.
Comment #18. Line 274: Does DST exist as a hydrate?
Response: We used monohydrate form of DST in the work. It is mentioned in the manuscript (Line 70).
Comment #19: Line 277: The authors think that the percentage of DST was so low that it could not be detected?
Response: PM-5 contains 16.7% drug compared PM-1. Furthermore, sample size is also low in DSC (1-2 mg). That is why we could not see crystalline DST peak in PM-5.
Comment #20. Line 279: Actually a peak seems t be present at about 260°C, which could be de depressed melting point of the drug actually, moreover also the water peak of DST is present. Do the authors think actually they might have a mixture of a new polymorph with the original one? Also in the XRPD, some peaks of the recrystallized DST are in line with the original polymorph.
Response: We have modified the DSC section of ‘results and discusion’ as per viwer comments.
Comment #21. Line 281: the reviewer thinks there is a very small peak present at 240°C, again this could be the depressed peak of the original polymorph.
Response: We have modified the DSC section of ‘results and discusion’ as per viwer comments (Line 354).
Comment #22. Line 284: Please be more specific about how the DSC results are in line with XRPD ones, how do the results connect? XRPD and DSC are complementary techniques and thus the reviewer thinks that the manuscript would benefit if the authors join the 2 techniques and discuss them together.
Response: XRPD and DSC supported amorphous phase transformation in the initial sample and recrystallization in the stability sample. We have modified the DSC section of ‘results and discusion’ section as per viwer comments.
Comment #23. Line 284: why did the authors not discuss the results after stability? For instance ASD-1 just the water peak is clearly visible now, why? For ASD-5 the peak at 240°C seems now more preeminent.
Response: We have modified the DSC section of ‘results and discusion’ as per viwer comments. Also, combined initial and stability data in the ‘results and discusion’ section of the manuscript to enhance clarity of data.
Comment #24. Line 284: why did the authors not analyze Tg for the system or the miscibility of the components via melting peak depression in the PM? It would be good to know how the drug is a polymer mix as this is a good predictor of the stability of systems. Moreover, this could also help understand why the polymer did not allow the full amorphization of the drug (poor miscibility some drug-rich zones in the drug could be less stable and crystallinity is promoted, for instance).
Response: Our DSC equipment does not have modulated option. However, we added estimated Tg information in the manuscript (Line 253 and Table 1).
Comment #25. Line 302: were these the nominal expected values? Are these within the accepted nominal dose ranges established by compendial guidelines? Moreover, why did the authors here analyze all ASDs and before only ASD-1 and ASD-5?
Response: These are nominal values. Yes, these are within the accepted range as these values were within range. Please see response to comment #4.
Comment 25. Line: 324: the authors describe well the effect of moisture uptake on the Tg and stability, thus why was the Tg not evaluated and the moisture content also?
Response: Please see the response of comment #24.
Comment 26. Line 379: how do the authors know that the Tg of ASD-5 was higher? They did not evaluate the Tg via DSC. What is the Tg of the drug, and what is the Tg of the polymer? How do the authors know the polymer has a higher Tg than the drug?
Response: Please see the response of comment #24.
Comment 27. Line 380: please avoid word shortening such as it’s
Response: Thank you pointing out typo error. It is corrected in the manuscript (Line 255).
Comment 28. Line 390: if there was a transformation to the original polymorphic form, why was no peak detected at 280°C?
Response: We have modified the DSC section of ‘results and discusion’ as per viwer comments.
Comment 29. Line 363: by the analysis of the DSC the reviewer thinks there is a peak at 240°C for the ASD-5 and this peak is more evident after stability, this could appear insignificant, but could indicate already some conversion of ASD-5 after 1 week at accelerated conditions that could explain the small difference in dissolution, although, for the results of this study this was not very relevant, it could raise some concerns about the long term shelf stability of the formulations and in-use performance.
Response: We have modified the DSC section of ‘results and discusion’ as per viwer comments. This is proof of concept data. The stability studies was performed in a scintillation vial not a HDPE bottle. Formulation optimization need additional work to further improve dissolution and stability.
Comment 30. Line 436: could the authors please add the absolute bioavailability, likewise it would be more straightforward to compare with already marketed formulations that are only 14-34% bioavailable.
Response: Absolute bioavailability of dasatinib is not reported in human due to lack of intravenous formulation.
Comment 31. Line 437: could the authors please state the Cmin of the Cp v. Time profiles?
Response: Cmin values of 2.45 and 3.10 ng/ml were observed at 24 h in PM and ASD formulation, respectively. It is added in the manuscript (Line 443).
Round 2
Reviewer 1 Report
The authors have modified the manuscript as per suggestions and the manuscript is ready to accept.
Reviewer 2 Report
The authors addressed the reviewer's comments satisfactorily and is a good initial study on the ASD of a very important class of compounds.